# Value Orientations, Personal Norms, and Public Attitude toward SDGs

**DOI:** 10.3390/ijerph20054031

**Published:** 2023-02-24

**Authors:** Ting Guan, Qian Zhang

**Affiliations:** School of Government, Beijing Normal University, Beijing 100875, China

**Keywords:** public attitude, value orientations, personal norms, SDGs

## Abstract

Improving communication and engagement with the public is vital for implementing sustainable development goals (SDGs). Public attitude toward SDGs can influence this engagement, as people are more likely to accept SDG-relevant information and take actions that are consistent with their own attitudes. This study examines the determinants for individual attitudes in supporting SDGs and further explores the formation of public attitudes toward SDGs, i.e., how public attitude is shaped by the value orientations and norms of the individuals. Using an online survey (n = 3089), we uncovered several important findings: (1) individuals’ altruistic/biospheric value orientations are positively associated with pro-SDG attitudes; (2) personal norms mediate the relationship of individuals’ altruistic values and attitudes; (3) some demographic characteristics (i.e., age, gender, having children) moderate the relationship of people’s value orientations and attitudes; and (4) people’s biospheric values have heterogeneous effects on their pro-SDG attitudes based on education and income. Through these findings, this study enhanced the public’s general understanding of SDGs by providing a holistic analytical framework of public attitude formation on SDGs and uncovering the significant role of value orientations. We further identify the moderating effects of demographic characteristics and the mediating effects of personal norms in the relationship between individuals’ values and attitude on SDGs.

## 1. Introduction

Sustainable development (SD) aims to advance our civilization in an environmentally, socially, and economically sustainable way to ensure a prosperous future for ourselves and future generations [1,2]. To address global challenges and threats, the United Nations set 17 sustainable development goals (SDGs) for 2030 and offered an extensive framework for guiding global efforts toward human prosperity in the long term [3]. It is encouraged that the SDGs be implemented through multi-stakeholder engagement, partnership, and voluntary commitments [4]. Achieving the SDGs requires that all countries, relevant stakeholders, and individuals work together in a synthesized way; there is also a critical need to mobilize and involve the public to find solutions to SDG-related challenges. Promoting public support not only increases policy acceptance and active participation in SDG-relevant programs but also increases public pressure on authorities, businesses, and third parties to take actions. New technological advancements have begun to enable innovations in tackling challenging problems, i.e., citizens increasingly developing new SD measures and ideas through social media, crowdsourcing, and open-source databases [5,6,7,8]. Therefore, it is vital to gain a better understanding of the determinants and formation of public attitude and behaviors toward SDGs.

Despite the first rule of effective communication—“know your audience”, studies on public attitude toward SDGs have been limited. Several studies have adopted the perspective of policymakers in understanding and taking actions on SDGs [9], yet little is known about value orientations and public attitude toward SDGs. In fact, other than the major environmental issues that extend beyond the topic of SDGs, our understanding of public views on SDGs is limited at best [10]. To better promote public communication and build public support for SDG policy responses, a thorough analysis of public attitude toward SDGs is urgently needed [11,12].

Previous studies show that people’s attitudes are influenced by many factors, including personal values, norms, beliefs, knowledge of the topic, and prior experiences [13,14,15]. In this study, we examine the determinants of individual attitudes in supporting SDGs and further explore the formation of public attitude toward SDGs, i.e., how public attitude is shaped by the value orientations and norms of the individuals. This study develops our understanding of public views of SDGs in the following ways. First, we integrate previous attitudinal models and findings and provide a holistic view of public attitude formation on SDGs. Second, we deepen our understanding of the role of value orientations in the formation of attitudes toward SDGs. Third, we further examine the mediating role of personal norms between an individual’s value orientations and pro-SDG attitudes. Specifically, we explore the moderating effects of personal characteristics on the relationship between people’s value orientations and their supportive attitudes toward SDGs.

## 2. Literature Review

### 2.1. Formation of Public Attitude

In the literature, the term “public attitude” usually suggests the feelings or evaluations of the public toward certain issues, such as pleasant or unpleasant, good or bad, and harmful or beneficial [16]. Eagly and Chaiken offer a widely applicable definition of attitude as “a psychological tendency that is expressed by evaluating a particular entity with some degree of favor or disfavor”; in this definition, they emphasize the features of evaluation, attitude object, and tendency [17]. Similarly, Breckler defines attitude as an individual’s response to external stimuli or specific subjects [18]. In this study, we follow the definition of public attitude as “the evaluation judgments that pertain to support among the public for particular issues” [19]. In addition, attitude can be reflected in one’s expression in support of or opposition to a certain view to varying degrees. Thus, we define public support as “public attitude that reflects the preferences and favorability among the public on certain issue” [19].

Numerous studies seek to unravel the nature of attitude formation. Maguire identifies key factors that influence one’s initially established attitudes, including “genetic determinants, transient physiological factors, direct experience with the attitude objects, and social processes like socialization, indoctrination, and peer pressure” [20]. Schwartz explores the role of altruistic norms in prosocial behavior, presuming that personal norms are based on an individual’s value orientations [21]. According to Schwartz’s norm-activation model, attitudes are generated from personal values, information about attitude objects, and social interactions [21,22]. Similarly, Stern et al. developed the value-belief-norm (VBN) theory, suggesting that individual attitudes are shaped by referencing value orientations and beliefs about the consequences of the objects for their values [23]. According to VBN theory, personal values serve as guiding principles in people’s lives and influence individuals’ worldviews (e.g., acceptance of New Ecological Paradigm), which are influenced by personal beliefs (awareness of consequences and ascription of responsibility) and personal norms [24].

### 2.2. Public Attitude toward SDGs

Addressing SDGs requires various stakeholders’ commitment and actions, including governments, companies, and non-profit organizations; in addition, the public plays an important role in promoting SDGs [4]. Previous studies show that individuals can act to promote SDGs through a shift in mindset, sustainable lifestyles, policy support, and civic engagement [25]. Although actions are easier to observe and quantify, the mindset is considered to be more fundamental and critical. In addition, since individual attitudes can predict and explain behaviors [26,27], it would be helpful to promote the public’s pro-SDG actions by better understanding public attitude toward SDGs. 

Previous studies can be categorized into two groups: studies of the public’s understanding of SDGs [10,11,12], and studies of the determinants and influential factors on public attitudes [19,28]. Some studies focus on the perceptions and understandings of the public about SDGs (public perception used here to describe how the public understands and thinks about the SDGs, i.e., people’s general impression about the SDGs). Research shows that people in different countries assign SDGs to economic/environmental/social pillars to varying degrees [10,12]. Bain et al. identify mental maps of people’s perceptions of SDGs which reveal perceived tensions among three aspects of SD in different countries [10]. Similarly, a global survey mapping out students’ perceptions of SDGs in 41 countries shows that students consider environmental, economic, and social challenges to have varying degrees of importance [12]. Interestingly, the same study suggests that SDGs as a whole is considered less important in wealthier countries and that students perceive the economic dimension as less important in SDGs [12]. Moreover, the public understands each SDG quite differently from the UN’s descriptions [11]. These studies reflect the deep gap that remains between public perceptions and the actual content of SDGs.

Other studies mainly focus on the determinants and influential factors on public attitude toward SDGs. Among these, cognitive variables, including individuals’ knowledge about SDGs and the education system/method, are considered to be important for enhancing the SDG’s actions [19,28]. In addition, social-psychological variables, including those factors related with personality characteristics, personal values, and worldviews, are believed to impact attitudes toward SDGs. Guan et al. show that Chinese people’s altruistic values and anthropocentric worldviews are positively related with their supportive attitudes toward SDGs [19]. By comparing students’ perceptions in different countries, Kleespies and Dierkes find that people in industrialized countries consider SDGs less important than people in developing countries [12]. In general, recent studies show that psychological factors and knowledge of individuals about SDGs strongly influence individuals’ engagement with SDGs [10,19].

However, gaps remain in understanding people’s attitude toward SDGs. First, studies on people’s pro-environmental attitudes and behaviors cannot directly predict people’s pro-SDG actions. Previous studies find that the public is prone to perceive SDGs as a general development phenomenon rather than distinct environmental issues [12,19]. A noteworthy finding is that postmaterialist values do not necessarily lead to higher support for SDGs [12,19]. This finding suggests that people perceive SDGs differently from the issue of environmental protection. Second, influential factors between personal values and public attitude still require further exploration. At the national level, Zheng et al. highlight the important linkages between cultural values and SDG achievements [29]. The role of culture, however, has been rarely addressed at the individual level on public attitude toward SDGs. As contemporary studies increasingly address culture as an endogenous variable, i.e., empirical advances in social science explore culture at the individual level by employing the value-belief-norms framework [30], it is necessary to explore the role of personal values, norms, and attitude on SDGs.

## 3. Analytical Framework and Hypotheses

### 3.1. Value Orientations and Individual Attitudes toward SDGs

As SDGs are perceived as a general development phenomenon, rather than distinct environmental or economic issues, it is important for the public to take a broader and more altruistic perspective in terms of actions. The concept of SD emphasizes holistic development in the environmental, economic, and social spheres [31], and its intellectual roots lie within the contradiction of capitalism between interlinked economic, social, and ecological crises. In essence, the core idea of SD relates to the balance between human-made and natural capital [32]. To achieve human-nature balance, people’s altruistic value orientations are important because SD requires even more individuals’ principled actions for reasons other than self-interest.

Based on Schwarz’s norm-activation model [21], attitude derives from human values, beliefs, and personal moral norms which are based on altruism [33]. Individuals’ altruistic values and beliefs, based on VBN theory, are closely related to personal norms, which lead to altruistic intentions and behaviors, such as helping, sharing, and pro-social and pro-environmental behavior [24]. In other words, an individual’s attitude toward sustainability could flow from his/her value orientations that reflect their concerns for other human beings and nonhuman species [34]. Accordingly, we propose that altruistic values and personal norms are important for the public to form supportive attitudes toward SDGs.

Previous studies suggest categorizing individuals’ value orientations into three types: egoistic value orientation, altruistic value orientation, and biospheric value orientation [35]. Practices based on different value orientations have varied emphases: practices based on altruistic value orientations emphasize altruism toward fellow humans, those based on biospheric value orientations emphasize the welfare of nonhuman species or the biosphere itself, while those based on egoistic values emphasize the importance of self-interests. Concerning the social and environmental pillars of SDGs, people’s altruistic values serve a key role in promoting civil rights and social justice [24], while biospheric values are important for improving environmental performance and biological diversity. Previous research verifies the influential role of altruistic/biospheric value orientations on public concerns with pro-environmental actions and support for social movements (e.g., [23,24]). These factors have not yet been explored empirically in terms of public attitude toward SDGs. Accordingly, we propose the following hypotheses:

 **Hypothesis 1a (H1a).** 
*Individuals with altruistic value orientations will be more supportive of SDGs than those with egoistic value orientations.*


 **Hypothesis 1b (H1b).** 
*Individuals with biospheric value orientations will be more supportive of SDGs than those with egoistic value orientations.*


### 3.2. The Mediating Role of Personal Norms in Value-Attitude Relations

Personal norms refer to individuals’ internalized norms which consist of personal expectations, obligations, and sanctions to certain behaviors that are anchored in the self [21]. They are feelings of personal obligation that compel individuals to act in ways that support certain beliefs (referred to as “activation of self-expectations”) [21,24]. The formation of norm orientations is influenced by individuals’ psychological predispositions, social interactions, knowledge construction, and information processing. The norm-activation model suggests that the variation in attitude toward certain objects is rooted in individuals’ universal values and is generated from personal norms related to the objects [24]. Therefore, individuals’ attitudes toward SDGs are likely to be generated when they hold certain expectations on the issue of SD. In short, individuals’ attitudes toward SDGs not only stem from value orientations but are also influenced by personal norms in regard to SD; in addition, personal norms in regard to SD are likely to mediate value-attitude relations. Accordingly, we propose the following hypotheses:

 **Hypothesis 2a (H2a).** 
*Personal norms related to SD mediate the impact of altruistic value orientations on individuals’ attitudes toward SDGs.*


 **Hypothesis 2b (H2b).** 
*Personal norms related to SD mediate the impact of biospheric value orientations on individuals’ attitudes toward SDGs.*


### 3.3. The Moderating Role of Personal Characteristics in Value-Attitude Relations

Previous studies show that gender, age, education, political values, cultural traits, personal values, and knowledge about SDGs are highly related to public support of SDGs. Similarly, socio-demographic variables, including gender, age, and education are found to exert an impact on people’s pro-SDG attitudes. Fløttum et al. conducted an empirical investigation of Norwegian citizens and found that young, female respondents with higher education typically hold more positive views on SDGs [36]. These results agree with the findings from a similar survey of five cities in China [19]. Thus, in this study, we control age, gender, income, having children, level of education, and knowledge about SDGs as personal characteristics (see in Figure 1).

To better understand the influential mechanisms between value orientations and individuals’ pro-SDG attitudes, we further explore the interaction effects between personal characteristics and value orientations on his/her attitudes. As it has been empirically verified that age, gender, and education are closely related to supportive public attitude toward SDGs [19,36], we propose the following hypotheses:

 **Hypothesis 3a (H3a).** 
*Age moderates the effect of individuals’ altruistic/biospheric value orientations in regard to support for SDGs.*


 **Hypothesis 3b (H3b).** 
*Having children moderates the effect of individuals’ altruistic/biospheric value orientations in regard to support for SDGs.*


 **Hypothesis 3c (H3c).** 
*Education moderates the effect of individuals’ altruistic/biospheric value orientations in regard to support for SDGs.*


 **Hypothesis 3d (H3d).** 
*Gender moderates the effect of individuals’ altruistic/biospheric value orientations on support for SDGs.*


 **Hypothesis 3e (H3e).** 
*Income level moderates the effect of individuals’ altruistic/biospheric value orientations on support for SDGs.*


### 3.4. Analytical Framework and Hypotheses

Based on the above literature review and analysis, we propose three main hypotheses which are visualized in Figure 1. The first set of hypotheses (H1a & H1b) examines the relationship between value orientations and people’s attitudes toward SDGs. The second set of hypotheses (H2a and H2b) focuses on the role of personal norms in value-attitude relationships. The third set of hypotheses (H3a, H3b, H3c, H3d, H3e) examines the moderating effect of personal characteristics (age, gender, education, having children, and income level) on value-attitude relations.

The measurement of individual attitudes is significant because it is a latent attribute that cannot be directly and precisely measured by observable indicators. In fact, the theoretical postulated relations between observable indicators and latent attributes impose a first-order control on the measurements [37]. The most widely used attitude measurement model is the classical tripartite model, developed by Rosenberg and Hovland [38], which emphasizes cognitive, affective, and behavioral components of attitudes (see also, [17]). In this study, we follow the tripartite model to measure the cognitive, affective, and behavioral manifestations of individual attitudes toward SDGs. This measurement, however, has a limitation due to its weak relevance to overt behaviors [27,39,40]. We overcome this limitation by emphasizing the dimensions of both affective expression and behavioral intention [41].

## 4. Data and Methods

### 4.1. Data

To better understand public values, knowledge, and attitude toward SDGs, we conducted an online national survey in August 2020, supported by the Institute for Sustainable Development Goals, Tsinghua University (TUSDG). An online questionnaire was used to collect data through random distribution to pre-screened respondents in China in the sample pool. The questionnaire contained 36 questions primarily covering basic demographic data as well as respondents’ values, knowledge, attitudes, and expectations concerning SDGs. Seven pre-tests were carried out to ensure the reliability and accuracy of the questionnaire content. A total of 3089 individuals in 30 provinces returned valid questionnaire responses for our analysis. All participants provided written informed consent prior to enrollment in the study. The date and time were individually recorded for each of the respondents to ensure that the returned questionnaire was valid.

Our sample originated from the dataset that consisted of 3089 respondents between the ages of 12 and 89 (M = 30.02, SD = 7.61). Of the respondents, 57% are female (SD = 0.49); 60% have children (SD = 0.49); 18% have less than or equivalent to an associate degree; 72% have a bachelor’s degree; 10% have a master’s degree and above (M = 1.92, SD = 0.52); 9% are low income with a monthly salary below 2000 RMB; 19% are middle income (2000–5000 RMB); 44% are mid-high income (5000–10,000 RMB); and 29% are high income (above 10,000 RMB) (M = 2.93, SD = 0.90). Table 1 lists the descriptive statistics for all variables.

### 4.2. Measures

The key explanatory variables in this study include knowledge of SDGs, altruistic values, and personal norms. Knowledge of SDGs was determined using the answers to four basic questions regarding SDGs, including the originator of SDGs, the number of SDGs, the logo of SDGs, and the relationship between SDGs and the 2030 Agenda (i.e., Transforming our world: the 2030 Agenda for Sustainable Development). Each correct answer was counted as one point for the knowledge of SDGs (M = 1.78, SD = 1.37). Concerning the measurement of personal value orientations: altruistic value orientations were operationalized by the respondent’s willingness to participate in the voluntary actions of social organizations (1 = very unlikely, 5 = very likely, M = 3.76, SD = 0.79), emphasizing altruism toward fellow humans, and biospheric value orientations were operationalized by whether the respondent would feed stray animals (cats, dogs, etc.) on the street (1 = no, 2 = sometimes, 3 = often, M = 1.93, SD = 0.59), focusing on the welfare of nonhuman species. Personal norms were measured by the level of support for the statement “Our society should not only focus on economic development but should pay more attention to the balanced development of economy, society and environment” (1 = very unsupportive, 5 = very supportive, M = 4.41, SD = 0.80).

The dependent variables related to individual attitudes were measured in two ways: attitudes toward SDGs were operationalized by asking the respondent’s attitudes toward the UN setting SDGs (1 = very unsupportive, 5 = very supportive, M = 4.11, SD = 0.75), and behavioral intention of SDGs was operationalized by willingness to donate financially to an SDG volunteer action project initiated by a company or community, ranging from zero to over 500 in RMB (M = 3.26, SD = 0.86). Table 2 summarizes the measurement of variables.

## 5. Results

All analyses in this study were conducted using Stata 15. We report results in three dimensions. In the first section, we perform a multivariate regression analysis and report results on determinants of individual attitudes and behavioral intention. In the second section, we conduct a mediation analysis and show results on the role of personal norms in altruistic/biospheric value orientations and pro-SDG attitudes. In the third section, we examine the moderation and heterogeneous effects and present results on the role of personal characteristics in the relationship between value orientations and pro-SDG attitude.

### 5.1. Determinants of Public Attitudes and Behavioral Intention

#### 5.1.1. Attitudes

The results of the multivariate regression analysis are reported in Table 3. The first model shows how attitudes are formed based on demographic characteristics; female respondents with higher educational levels, low income, and more knowledge of SDGs are significantly more likely to hold pro-SDG attitudes. The second model incorporates altruistic/biospheric values; respondents with altruistic values, higher education levels, and more knowledge of SDGs are significantly more inclined to have pro-SDG attitudes.

#### 5.1.2. Behavioral Intention

Table 4 shows the results for the determinants of behavioral intention. The first model reveals how demographic variables predict behavioral intention. Respondents who are younger, hold a bachelor’s degree, have children, have higher income, and have more knowledge of SDGs are significantly more likely to display behavioral intention in favor of SDGs. The second model indicates how altruistic/biospheric values predict behavioral intention, controlling for demographics. Younger respondents with altruistic and biospheric value orientations who have children, higher income, and more knowledge of SDGs are significantly more inclined to exhibit behavioral intention for SDGs.

These results show that determinants of attitudes differ from determinants of behavioral intention. Among demographic characteristics, only a higher educational level is a significant predictor of attitudes, while lower age, having children, and higher income significantly predict behavioral intention. More knowledge of SDGs is a significant determinant of both attitudes and behavioral intention. In support of Hypothesis 1a, people with altruistic value orientations are more supportive of SDGs, both in attitudes and behavioral intentions, than are those with egoistic values. In support of Hypothesis 1b, people with biospheric value orientations are more supportive of SDGs in behavioral intention than are those with egoistic values.

### 5.2. Role of Personal Norms

Building on Hypothesis 1, this study further examines the role of personal norms in the relationship between altruistic/biospheric values and attitudes. Table 5 presents the bootstrap results. The first model shows the mediation analysis results, controlling for demographics and knowledge of SDGs. The direct effect of altruistic values on attitudes is 0.220 (*p* < 0.001) and the indirect effect through personal norms is 0.049 (*p* < 0.001). The mediation effect of personal norms explains 18.22% of the total effect. Figure 2 provides visualizations of the mediation analysis results.

The results of the mediation analysis support Hypothesis 2a, suggesting personal norms related to SD mediate the impact of altruistic values on individuals’ attitude toward SDGs. Hypothesis 2b is not supported, however, as the results suggest personal norms related to SD do not mediate the impact of biospheric values on individuals’ attitudes toward SDGs. These results are consistent with intuition since biospheric values are not significant predictors of attitudes. Furthermore, the results demonstrate that the formation of personal norms is less significantly related to behavioral intention compared to attitudes.

### 5.3. Moderating Effect of Demographics

#### 5.3.1. Age

We found that interactions between age and altruistic values to be significant for attitudes toward SDGs (Table 6). (Control variables included age, gender, education, having children, income and knowledge of SDGs in Table 6, Table 7, Table 8, Table 9, Table 10 and Table 11.) The first model suggests that age moderates the relationship between altruistic value orientations and attitudes. The moderation effect is visualized in Figure 3, showing the slopes of the average age (±1 standard deviation). As respondents are more prone to altruistic value orientations, the effect becomes stronger for older respondents than for younger respondents on pro-SDG attitudes. It should be noted that the variable altruistic value is not significant in the first model, suggesting the relationship between altruistic values and attitudes is not significant when age is zero, which is considered an invalid result for the sample. The second model finds no such moderation effect for biospheric value orientations and attitudes; the effect of biospheric values is found to be the same for respondents of all ages.

A similar moderation effect of age is found for altruistic values and behavioral intention toward SDGs (Table 7). The first model indicates that age significantly moderates the relationship between altruistic values and behavioral intention; an interaction plot is displayed in Figure 4. As altruistic values become stronger, the effect for older rather than younger respondents on the behavioral intention of SDGs becomes more significant. The second model shows no such interaction effect, revealing no heterogeneous effect of biospheric value orientations by age on behavioral intention.

#### 5.3.2. Having Children

To test the moderation effect of having children, we include an interaction between having children and values orientations (altruistic/biospheric values) in each model (Table 8). The first model demonstrates having children moderates the relationship between altruistic values and attitudes; the interaction plot is displayed in Figure 5. As respondent’s altruistic values increase, the effect becomes stronger for respondents who have children than for those without children on pro-SDG attitudes. The second model reveals no such moderation effect for biospheric values and attitudes, however; the effect of biospheric values is found to be the same for respondents with and without children.

#### 5.3.3. Education

Following the same approach, we found no significant interaction effect between education and altruistic values on attitudes. However, we found a heterogeneous effect of education by running separate OLS regression models on three education levels (Table 9). Biospheric values are significantly positively related to pro-SDG attitudes for respondents with a bachelor’s or a master’s degree and above, while no such relationship is found for respondents holding less than or equivalent to an associate degree. The coefficients by education level are visualized in Figure 6.

In support of Hypothesis 3b, the results suggest that biospheric values are a significant predictor of pro-SDG attitudes only for respondents with higher education levels. For those with a bachelor’s or a master’s degree and above, stronger biospheric values correlate to higher support of SDGs. For those with less than or equivalent to an associate degree, biospheric values are not significantly related to pro-SDG attitudes. The non-significant relationship between biospheric values and attitudes for respondents holding less than or equivalent to an associate degree may explain the finding of a non-significant interaction effect.

#### 5.3.4. Gender

A significant moderation effect for gender is found in the relationship between altruistic values and attitudes in the first model (as shown in Table 10). The interaction plot is displayed in Figure 7. As altruistic values increase, the effect is greater for men than women on pro-SDG attitudes. In support of Hypothesis 3c, the results show that gender moderates the effect of individuals’ altruistic values on support for SDGs. The second model shows no significant moderation effect for gender and biospheric values, indicating the relationship between biospheric values and attitudes is the same for men and women.

#### 5.3.5. Income

Rather than an interaction effect, we found a significant heterogeneous effect of income in the relationship between biospheric values and attitudes. By running separate OLS regression models by income levels, we find a significant relationship between biospheric values and attitudes for low-income and high-income groups only, while such a relationship is not observed for middle and mid-high-income groups (Table 11). For the low and high-income groups only, respondents holding biospheric values are more inclined to support SDGs. The coefficients by income levels are visualized in Figure 8.

## 6. Conclusions

The main goal of this paper is to investigate the determinants of individual attitude in supporting SDGs and, in particular, how public attitudes are shaped by their personal value orientations and norms. As hypothesized, our findings show that individuals’ altruistic/biospheric values are positively associated with pro-SDG attitudes (H1a, 1b). We also find that this value-attitude relationship is mediated by individuals’ personal norms on SD (H2a). We further demonstrate the moderating role of some demographic characteristics (i.e., age, gender, having children) in the relationship between individuals’ value orientations and a supportive attitude toward SDGs (H3a, 3b, 3d). In addition, for some demographic characteristics (i.e., education level, income), heterogeneous effects were identified in the relationship between individuals’ biospheric values and pro-SDG attitudes (H3c, 3e). Our findings are consistent with previous research regarding determinants of individual’s attitude toward SDGs, factors of people’s behavioral intention on SDGs. We further identified the mediation effect of personal norms on altruistic values and attitude toward SDGs, and the moderation effect of demographic characteristics in the relationship of values and attitudes.

Understanding public attitude toward SDGs is important for effectively communication with the public regarding its willingness to support SDGs-relevant actions [10]. Previous studies show that sustainable advertising, individuals’ predispositions, and knowledge about SDGs influence public support for SDGs [19,42]. Our findings relating to value orientations, SDG-relevant knowledge, and attitudes are consistent with those of previous empirical studies [19,28] that suggest that people with more SDG-relevant knowledge and altruistic values are more likely to be supportive of SDGs. Compared with previous studies, our study not only provides a more holistic view to better understand the formation of public attitude (affective expression and behavioral intention), it also importantly deepens the general understanding of the role of altruistic/biospheric value orientations and personal norms in attitudinal formation toward SDGs.

Existing literature in the environmental field emphasizes the significant role of altruistic values on people’s pro-environmental behaviors [43], yet the relationship between individuals’ value orientations and their attitudes on SD, which encompasses a holistic view of environmental, economic, and social spheres, has not been empirically explored. This issue is extremely important, as human development is required to transition from a traditional economic-centric path onto a sustainable trajectory. Therefore, our study provides a preliminary exploration to better understand people’s altruistic values and their attitude toward SD.

Our study provides a better understanding of people’s pro-SDG behavioral intention by improving relevant measurements. First, it is observed that the measurement of individuals’ pro-SDG attitudes in previous studies has limitations due to its potential weak relevance to overt behaviors. We attempt to overcome this limitation by measuring both individuals’ affective attitude and behavioral intentions on SDGs. In so doing, we find that, compared with people’s affective attitude, their behavioral intentions on SDGs are more correlated to their personal economic conditions. This finding agrees with the Campbell Paradigm [40] which highlights the parameter of behavioral costs in attitude-behavioral studies. Second, we measure individuals’ altruistic values from two perspectives: altruistic values and biospheric values. Our empirical findings suggest that individuals with more knowledge of SDGs and with altruistic value orientations and biospheric value orientations are significantly more prone to show behavioral intention toward SDGs. While altruistic and biospheric values are both significant in the relationship between values and behavioral intention, only altruistic values are significant in the value-attitude relationship. Moreover, we also find that younger respondents with children and higher income are significantly more likely to show higher behavioral intention toward SDGs. These findings have major implications for understanding people’s pro-SDG actions, which are further relevant in terms of public engagement in SDG practices.

One other interesting finding is that we verified the mediating role of personal norms in values and attitudes. Specifically, personal norms related to SD mediate the impact of altruistic value orientations on individuals’ attitudes toward SDGs. Our finding is consistent with a meta-analysis based on the norm-activation theory conducted more than 20 years ago [24]. As personal norms are influenced by many factors, including individuals’ beliefs, values, and other endogenous psychosocial factors, it can be challenging for public sectors to persuade individuals to change their views and lifestyles related to SD. Constant effort should be made to promote a public mindset toward sustainable practices. Our study also confirms the significant correlation between individuals’ knowledge about SDGs and their supportive attitude toward SDGs. Based on this finding, it can be considered that enhancing SDG-relevant education and individual involvement are essential for promoting public support toward SDGs.

Another important factor we identified in this study is the moderation effects of demographic characteristics on the relationship between value orientations and public attitudes. Previous studies have found that gender, age, education level, and having children influence individuals’ attitudes toward SDGs [19,36]. Our findings are consistent with these previous studies, and we further show the moderating and heterogeneous effects of some demographic characters in the relationship between individuals’ altruistic/biospheric values and their pro-SDG attitudes. Based on this finding, guiding different demographics of the population with tailed initiatives may increase overall support for SDGs. Targeted public communication initiatives, for example, seeking public support from families with children, might be more helpful in promoting people’s pro-SDG actions.

While there are many considerations regarding what and how to communicate with the public in promoting SDG involvement, our study establishes some benchmarks for better understanding people’s value orientations, norms and attitudes toward SDGs. Future studies could employ experimental and longitudinal study designs to further explore the causal relationships between people’s values and attitudes on SDGs. Some aspects of people’s knowledge and attitudes toward SDGs which have not yet been explored in detail, such as the role of demographics in moderating people’s knowledge–attitude relationships, could be potential topics for future research on promoting SDGs.

## Figures and Tables

**Figure 1 ijerph-20-04031-f001:**
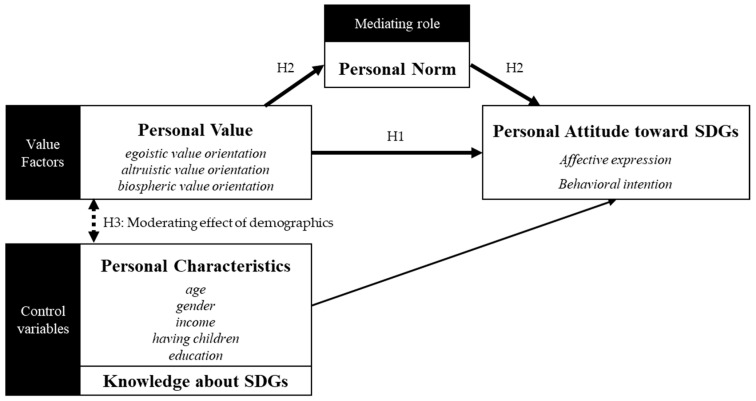
Personal value orientations, norms and individuals’ attitude toward SDGs.

**Figure 2 ijerph-20-04031-f002:**
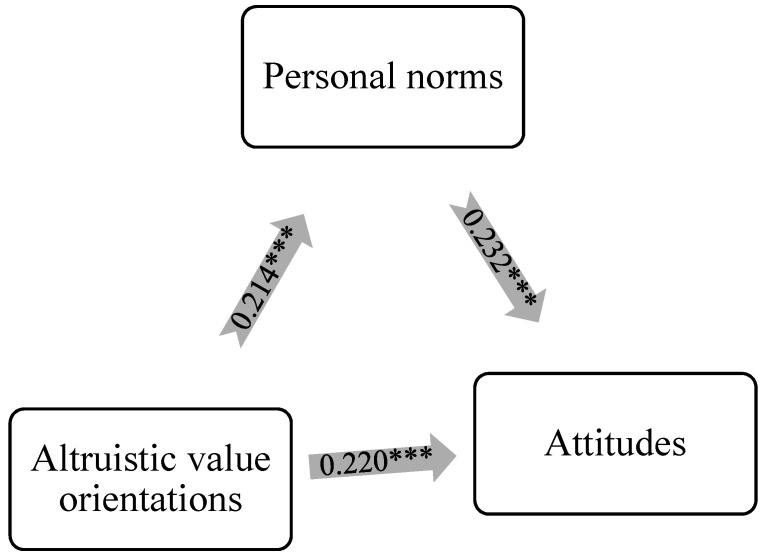
Mediation relationship between altruistic values, personal norms, and attitudes controlling for knowledge. Note: *** *p* < 0.001.

**Figure 3 ijerph-20-04031-f003:**
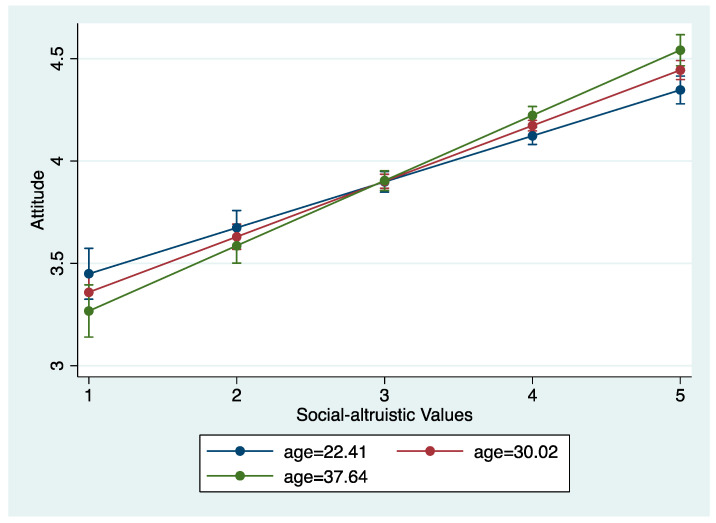
Interaction plot of age and altruistic values on attitudes.

**Figure 4 ijerph-20-04031-f004:**
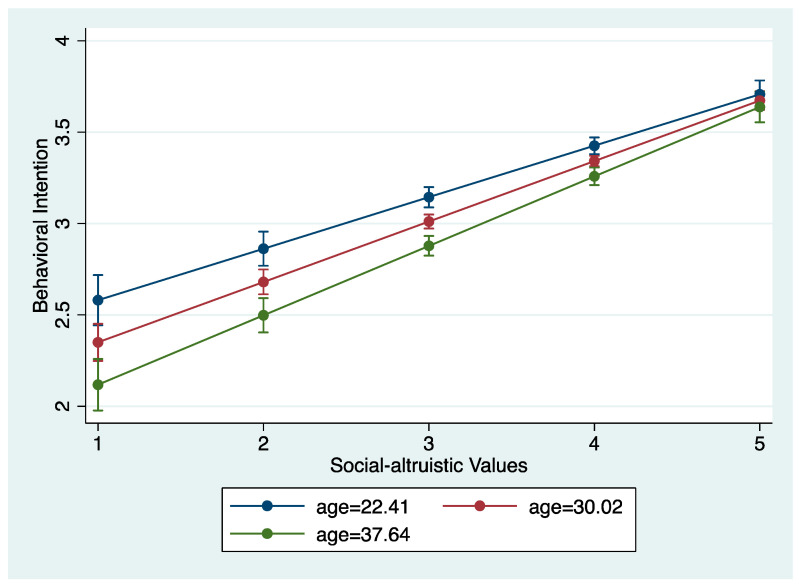
Interaction plot of age and altruistic values on behavioral intention.

**Figure 5 ijerph-20-04031-f005:**
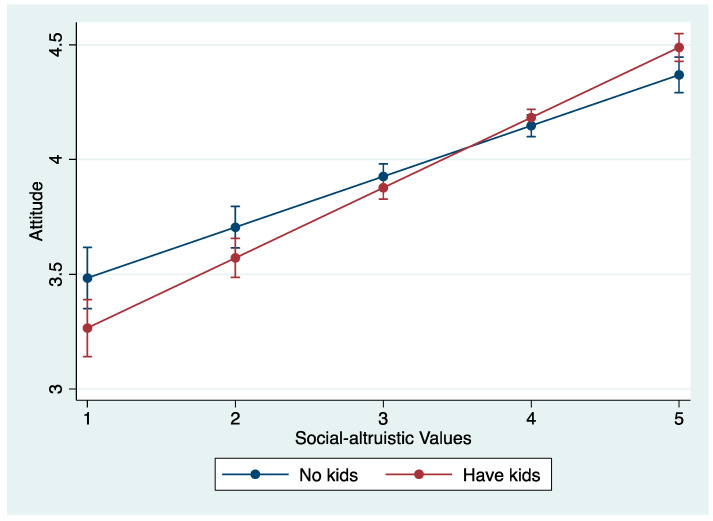
Interaction plot of having children and altruistic values on attitudes.

**Figure 6 ijerph-20-04031-f006:**
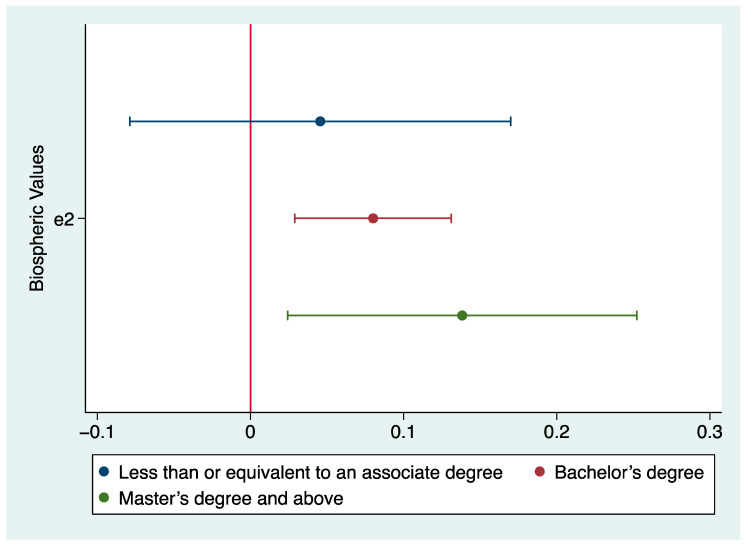
OLS coefficients of biospheric values on attitude by education, no covariates shown.

**Figure 7 ijerph-20-04031-f007:**
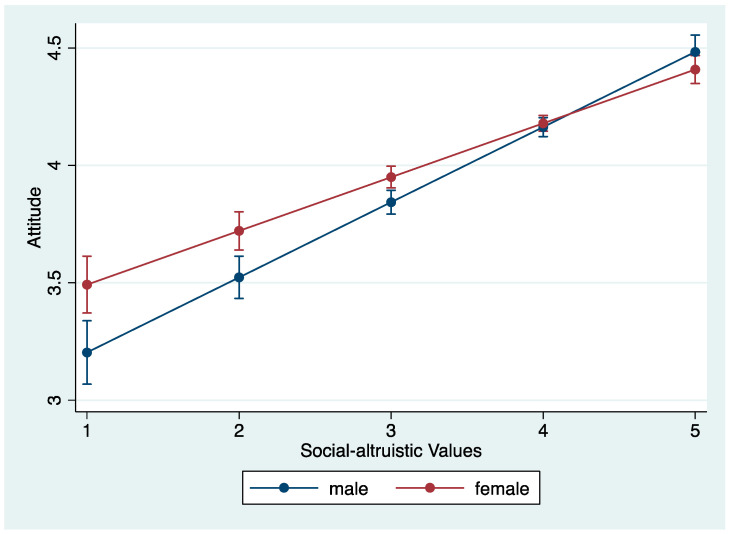
Interaction plot of gender and altruistic values on attitudes.

**Figure 8 ijerph-20-04031-f008:**
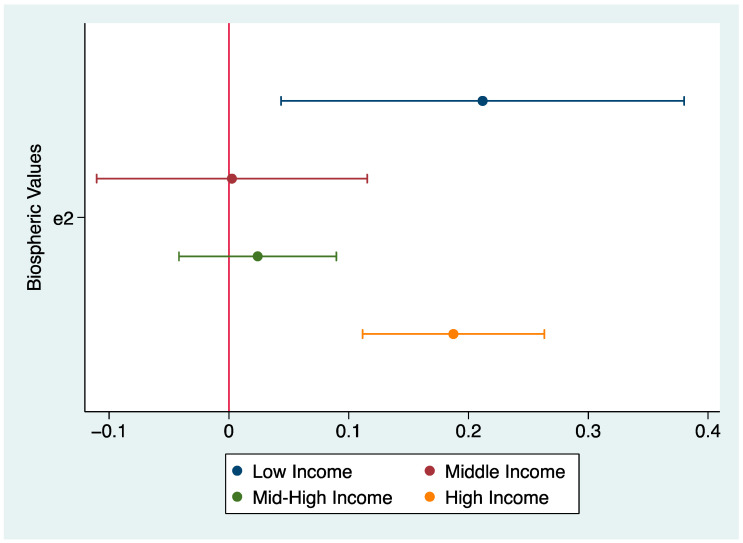
OLS coefficients for biospheric values on attitudes by income, no covariates shown.

**Table 1 ijerph-20-04031-t001:** Descriptive statistics for variables (n = 3089).

Variable	Mean	SD	Min	Max
Age	30.02	7.61	12	89
Gender	0.57	0.49	0	1
Having children	0.60	0.49	0	1
Education	1.92	0.52	1	3
Income	2.93	0.90	1	4
Knowledge of SDGs	1.78	1.37	0	4
Altruistic value orientations	3.76	0.79	1	5
Biospheric value orientations	1.93	0.59	1	3
Personal norms	4.41	0.80	1	5
Attitudes toward SDGs	4.11	0.75	1	5
Behavioral intention of SDGs	3.26	0.86	1	5

**Table 2 ijerph-20-04031-t002:** Measurement of instruments (n = 3089).

Variable	n	Percentage
Altruistic value orientations (Are you willing to participate in the voluntary actions of social organizations such as publicity activities for wildlife protection?)		
Very unlikely	25	0.81
Somewhat unlikely	101	3.27
Depends	969	31.37
Somewhat likely	1485	48.07
Very likely	509	16.48
Biospheric value orientations (Would you feed stray animals (cats, dogs, etc.) on the street?)		
No	646	20.91
Sometimes	2008	65.00
Often	435	14.08
Personal norms (Our society should not only focus on economic development but should pay more attention to the balanced development of economy, society, and environment.)		
Very unsupportive	14	0.45
Somewhat unsupportive	76	2.46
Supportive	292	9.45
Somewhat supportive	962	31.14
Very supportive	1745	56.49
Attitudes toward SDGs (What is your attitude toward the UN setting the SDGs?)		
Very unsupportive	20	0.65
Somewhat unsupportive	37	1.20
Supportive	496	16.06
Somewhat supportive	1578	51.08
Very supportive	958	31.01
Behavioral intention of SDGs (If your company or community initiates an SDG volunteer action project, how much in RMB are you willing to donate to this project?)		
0	105	3.40
0–10	327	10.59
10–100	1510	48.88
100–500	952	30.82
Above 500	195	6.31

**Table 3 ijerph-20-04031-t003:** Determinants of attitudes.

	Model 1			Model 2		
	Estimates	95% CI	*p*	Estimates	95% CI	*p*
Altruistic values				0.27	0.24, 0.30	**<0.001**
Biospheric values				−0.01	−0.05, 0.04	
Age	0.00	−0.00, 0.01	0.568	0.00	−0.00, 0.01	
Gender (Male)	0.06	0.01, 0.11	**0.023**	0.04	−0.01, 0.09	
Education (Less than or equivalent to an associate degree)						
Bachelor’s degree	0.12	0.05, 0.19	**0.001**	0.10	0.03, 0.17	**0.003**
Master’s degree and above	0.14	0.03, 0.25	**0.010**	0.12	0.02, 0.22	**0.021**
Having Children (No)	0.06	−0.00, 0.13	0.057	0.01	−0.05, 0.08	
Income (Low income)						
Middle income	−0.14	−0.25, −0.03	**0.013**	−0.10	−0.21, 0.00	0.053
Mid-High income	−0.10	−0.21, 0.00	0.054	−0.07	−0.17, 0.03	0.196
High income	−0.08	−0.20, 0.03	0.142	−0.09	−0.19, 0.02	0.112
Knowledge of SDGs	0.14	0.12, 0.16	**<0.001**	0.12	0.10, 0.13	**<0.001**
Constant	3.74	3.59, 3.89	**<0.001**	2.74	2.54, 2.94	**<0.001**
Observations	3089			3089		
R-squared	0.0839			0.1582		
Adj. R-squared	0.0812			0.1552		

Note: *p* < 0.05 in bold.

**Table 4 ijerph-20-04031-t004:** Determinants of behavioral intention.

	Model 1			Model 2		
	Estimates	95% CI	*p*	Estimates	95% CI	*p*
Altruistic values				0.28	0.24, 0.31	**<0.001**
Biospheric values				0.27	0.22, 0.31	**<0.001**
Age	−0.02	−0.02, −0.01	**<0.001**	−0.01	−0.01, −0.01	**<0.001**
Gender (Male)	0.01	−0.05, 0.07	0.650	−0.01	−0.06, 0.05	0.742
Education (Less than or equivalent to an associate degree)						
Bachelor’s degree	0.08	0.00, 0.16	**0.038**	0.07	−0.01, 0.14	0.068
Master’s degree and above	0.03	−0.09, 0.15	0.613	0.03	−0.09, 0.14	0.645
Having children (No)	0.29	0.21, 0.36	**<0.001**	0.19	0.12, 0.26	**<0.001**
Income (Low income)						
Middle income	0.19	0.07, 0.31	**0.003**	0.20	0.09, 0.32	**<0.001**
Mid-High income	0.35	0.24, 0.47	**<0.001**	0.36	0.26, 0.47	**<0.001**
High income	0.61	0.48, 0.73	**<0.001**	0.56	0.44, 0.67	**<0.001**
Knowledge of SDGs	0.10	0.08, 0.12	**<0.001**	0.06	0.04, 0.08	**<0.001**
Constant	2.99	2.82, 3.16	**<0.001**	1.38	1.16, 1.59	**<0.001**
Observations	3089			3089		
R-squared	0.1220			0.2371		
Adj. R-squared	0.1195			0.2344		

Note: *p* < 0.05 in bold.

**Table 5 ijerph-20-04031-t005:** Bootstrap mediation analysis of the role of personal norms.

	Model 1		
	Estimates	95% CI	*p*
Indirect effect	0.049	0.037, 0.062	**<0.001**
Direct effect	0.220	0.185, 0.255	**<0.001**
Total effect	0.269		
Percentage of mediation effect	18.22%		

Note: *p* < 0.05 in bold.

**Table 6 ijerph-20-04031-t006:** Moderating effect of age on the relationship between altruistic values and attitudes.

	Model 1			Model 2		
	Estimates	95% CI	*p*	Estimates	95% CI	*p*
Altruistic values	0.09	−0.04, 0.21	0.182			
Biospheric values				0.06	−0.12, 0.24	0.496
Age	−0.02	−0.03, −0.00	**0.021**	0.00	−0.01, 0.01	0.819
Age * Altruistic values	0.01	0.00, 0.01	**0.004**			
Age * Biospheric values				0.00	−0.01, 0.01	0.817
Controls	Yes			Yes		
Constant	3.40	2.91, 3.88	**<0.001**	3.60	3.24, 3.96	**<0.001**
Observations	3089			3089		
R-squared	0.1605			0.0878		
Adj. R-squared	0.1575			0.0845		

Note: * means interaction. *p* < 0.05 in bold.

**Table 7 ijerph-20-04031-t007:** Moderating effect of age on the relationship between altruistic values and behavioral intention.

	Model 1			Model 2		
	Estimates	95% CI	*p*	Estimates	95% CI	*p*
Altruistic values	0.14	−0.00, 0.28	0.055			
Biospheric values				0.32	0.13, 0.51	**0.001**
Age	−0.04	−0.05, 0.02	**<0.001**	−0.01	−0.03, −0.00	**0.023**
Age * Altruistic values	0.01	0.00, 0.01	**0.006**			
Age * Biospheric values				0.00	−0.00, 0.01	0.701
Control Variables	Yes			Yes		
Constant	2.46	1.92, 2.99	**<0.001**	2.29	1.90, 2.68	**<0.001**
Observations	3089			3089		
R-squared	0.2093			0.1793		
Adj. R-squared	0.2065			0.1764		

Note: * means interaction. *p* < 0.05 in bold.

**Table 8 ijerph-20-04031-t008:** Moderating effect of having children on the relationship between altruistic values and attitudes.

	Model 1			Model 2		
	Estimates	95% CI	*p*	Estimates	95% CI	*p*
Altruistic values	0.22	0.17, 0.27	**<0.001**			
Biospheric values				0.08	0.01, 0.15	**0.017**
Having children (No)	−0.30	−0.54, −0.06	**0.014**	0.06	−0.13, 0.24	0.548
Having children * Altruistic values	0.08	0.02, 0.15	**0.008**			
Having children * Biospheric values				−0.00	−0.09, 0.09	0.950
Control Variables	Yes			Yes		
Constant	2.91	2.68, 3.14	**<0.001**	3.56	3.35, 3.76	**<0.001**
Observations	3089			3089		
R-squared	0.1601			0.0878		
Adj. R-squared	0.1571			0.0845		

Note: * means interaction. *p* < 0.05 in bold.

**Table 9 ijerph-20-04031-t009:** Heterogeneous effect of education in the relationship of biospheric values and attitudes.

	Model 1			Model 2			Model 3		
	Less Than or Equivalent to an Associate Degree			Bachelor’s Degree			Master’s Degree and Above		
	Estimates	95% CI	*p*	Estimates	95% CI	*p*	Estimates	95% CI	*p*
Biospheric values	0.05	−0.08, 0.17	0.471	0.08	0.03, 0.13	**0.002**	0.14	0.02, 0.25	**0.018**
Control Variables	Yes			Yes			Yes		
Constant	3.51	3.12, 3.90	**<0.001**	3.71	3.51, 3.92	**<0.001**	3.65	3.16, 4.15	**<0.001**
Observations	555			2224			310		
R-squared	0.0924			0.0813			0.0690		
Adj. R-squared	0.0791			0.0779			0.0442		

Note: *p* < 0.05 in bold.

**Table 10 ijerph-20-04031-t010:** Moderating effect of gender in the relationship of altruistic values and attitudes.

	Model 1			Model 2		
	Estimates	95% CI	*p*	Estimates	95% CI	*p*
Altruistic values	0.32	0.27, 0.37	**<0.001**			
Biospheric values				0.08	0.01, 0.15	**0.021**
Gender (Male)	0.38	0.14, 0.62	**0.002**	0.05	−0.12, 0.23	0.551
Gender * Altruistic values	−0.09	−0.15, −0.03	**0.004**			
Gender * Biospheric values				0.00	−0.08, 0.09	0.934
Controls	Yes			Yes		
Constant	2.55	2.32, 2.78	**<0.001**	3.56	3.36, 3.77	**<0.001**
Observations	3089			3089		
R-squared	0.1605			0.0878		
Adj. R-squared	0.1575			0.0845		

Note: * means interaction. *p* < 0.05 in bold.

**Table 11 ijerph-20-04031-t011:** Heterogeneous effect of income in the relationship of biospheric values and attitudes.

	Model 1 Low Income			Model 2 Middle Income		
	Estimates	95% CI	*p*	Estimates	95% CI	*p*
Biospheric values	0.21	0.04, 0.38	**0.014**	0.00	−0.11, 0.12	0.966
Control Variables	Yes			Yes		
Constant	3.49	2.87, 4.11	**<** **0.001**	3.64	3.24, 4.05	**<** **0.001**
Observations	266			579		
R-squared	0.0717			0.0895		
Adj. R-squared	0.0465			0.0783		
	**Model 3** **Mid-High Income**			**Model 4** **High Income**		
	**Estimates**	**95% CI**	** *p* **	**Estimates**	**95% CI**	** *p* **
Biospheric values	0.02	−0.04, 0.09	0.474	0.19	0.22, 0.26	**<0.001**
Control Variables	Yes			Yes		
Constant	3.53	3.25, 3.81	**<0.001**	3.18	2.83, 3.53	**<0.001**
Observations	1347			897		
R-squared	0.0786			0.1232		
Adj. R-squared	0.0738			0.1163		

Note: *p* < 0.05 in bold.

## Data Availability

Data sharing not applicable.

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
