# Peer review of "Value Orientations, Personal Norms, and Public Attitude toward SDGs"

_ijerph, 2023, doi:10.3390/ijerph20054031_

Round 1

Reviewer 1 Report

Public attitudes are important in achieving SDG goals, and this paper touches on an important topic. The authors use online questionnaires to investigate the heterogeneity of public attitudes in the process of implementing the SDG goals in China and reveal the mediating effect of personal norms in the relationship between altruistic values and public attitude. The paper is well written and I have some questions or recommendations as follow. 

1. Individual's attitude and behavioral intention are both mentioned as dependent variables, so, what are the differences between these two factors? Are there any causal relationships between them in this study? 

2. The variable kid is unclear (is it number of children in the household?).

3. As SDG-relevant knowledge is also making a significant impact on individual's attitude on SDGs, I think it could be interesting to explore the relevance between individual's knowledge and attitude toward the SDGs, perhaps in another article. 

4. It is worth to add the discussion part which should take into account reference of results obtained in a given study to other research occurring in the literature (partly already included in conclusions). 

Author Response

Please see the attachment. Many thanks!

Reviewer 2 Report

The paper aims at a very interesting topic and it has been a real pleasure to read the manuscript. Nevertheless, I have some concerns to report:

1.- I feel that the title given to the draft paper seems tricky. It is a bit confusing as only altruistic values and personal norms are mentioned but the paper as well deals with personal/demographic characteristics and with biospheric values. According to what has just been mentioned, keywords need also to be reviewed. And the structure and sections need to be reconsidered (1.2 relates to altruistic values while there is not a separate treatment of biospheric values).

At some points, it seems that authors’ perspective is that biospheric values are somehow included in altruistic values (text on lines 265-266 and 270-275 may suggest so; text on lines 457-458 clearly stated that). If that is the case, then a clarification is vital in the whole section 1, as thereby the information is presented as if they were two different topics in a “personal value” category (see Figure 1 and lines 475-476). The lack of consistency makes the paper difficult to follow the introduction in its present version.

Also in order to improve the development and clarity of the first part of paper, it would be useful to deep into the notions of “public support”, “public behavior”, “people’s attitude” and “people’s perception” which are prolifically used in sections 1 and 1.1 without previous definition as if they were naturally comprehensible. After that, a “public attitude” definition is envisaged in section 1.2 (1) but the differences with the previous expressions are not clarified.

2.- Affirmation on lines 71-74 needs to be supported by a reference.

3.- Figure 1 includes in the “personal characteristics” box a mention to “knowledge about the SDGs” that has not been previously introduced when presenting what is going to be explored and the hypotheses proposed (1.2). As authors themselves indicate that the Figure visualizes the hypotheses (lines 223-223), the boxes should not include new items.

4.- I have already mentioned how confusing is the way in which the paper manages the references to altruistic values/social-altruistic values/biospheric values. I also want to point out that the items proposed in the research method to measure them (lines 270-275 & Table 2) leave a lot of room for improvement. On the one hand, because the example provided to answer the question on social-altruistic has itself a biospheric implication (wildlife); on the other hand, because the answers to the question regarding biospheric values are not so simple to interpret and measure: is feeding stray animals good or bad for biosphere? The personal/social characteristics of the respondent are very likely to affect that correlation and the feeding process needs to be sustainable and responsible in order to be considered positive. I cannot understand the choice of such blurry questions and the authors provide no information on that.

5.- The conclusion and discussion section needs to be reconsidered according to the previous remarks. One of the first conclusions is that “people with more SDG-relevant knowledge and altruistic values are more likely to be supportive of SDGs” (lines 456-454). That sentence involves two notions (knowledge about the SDGs & altruistic values) which are not clearly defined and presented in the paper. The same idea is on top of everything repeated soon after in the paper (lines 476-478) but it is presented in a slightly different way and making differences among social-altruistic values and biospheric values (lines 479-480).

Author Response

We would like to sincerely thank you for your thoughtful and valuable comments on the manuscript. Those comments are very helpful for us to improve our manuscript. We have revised our manuscript substantially based on the comments. The attached is our detailed response to your suggestions. We will explain how we addressed the issues raised in each of your comments one by one.

Round 2

Reviewer 2 Report

Authors have made a priceless effort to meet all my suggestions on the previous version. I think the paper has been truly improved and deserves publication, altough I still have some concerns regarding the questions chosen to measure altruistic and biospheric value orientations. In their cover letter, authors indicate that this measurment will be improved in future research actions. I am looking forward to hearing from them!